# Resilience as a Protective Factor in Basic Military Training, a Longitudinal Study of the Swiss Armed Forces

**DOI:** 10.3390/ijerph18116077

**Published:** 2021-06-04

**Authors:** Sandra Sefidan, Maria Pramstaller, Roberto La Marca, Thomas Wyss, Dena Sadeghi-Bahmani, Hubert Annen, Serge Brand

**Affiliations:** 1Department of Clinical Psychology and Psychotherapy, University of Zurich, 8050 Zurich, Switzerland; s.sefidan@hotmail.com (S.S.); maria@pramstaller.org (M.P.); Roberto.LaMarca@clinica-holistica.ch (R.L.M.); 2Military Academy, Swiss Federal Institute of Technology ETH Zurich, 8903 Birmensdorf, Switzerland; hubert.annen@milak.ethz.ch; 3Praxis Pramstaller, Seestrasse 107, 8707 Uetikon am See, Switzerland; 4Clinica Holistica Engiadina, Centre for Stress-Related Disorders, 7542 Susch, Switzerland; 5Swiss Federal Institute of Sport Magglingen SFISM, 2532 Magglingen, Switzerland; thomas.wyss@baspo.admin.ch; 6Sleep Disorders Research Center, Kermanshah University of Medical Sciences, Kermanshah 67146, Iran; dena.sadeghibahmani@upk.ch; 7Departments of Physical Therapy, University of Alabama at Birmingham, Birmingham, AL 35209, USA; 8Center for Affective, Stress and Sleep Disorders (ZASS), Psychiatric University Hospital Basel, 4002 Basel, Switzerland; 9Substance Abuse Prevention Research Center, Health, Institute, Kermanshah University of Medical Sciences, Kermanshah 67146, Iran; 10School of Medicine, Tehran University of Medical Sciences, Tehran 25529, Iran; 11Division of Sport Science and Psychosocial Health, Department of Sport, Exercise and Health, University of Basel, 4052 Basel, Switzerland

**Keywords:** resilience, perceived stress, mental distress, dropout, performance

## Abstract

For recruits, basic military training (BMT) can be experienced as a stressful episode in which relevant protective factors such as resilience might be essential for successful completion of the training. The present study examined whether resilience would act as a protective factor during BMT in the Swiss Armed Forces. To this end, we conducted a cross-sectional and longitudinal study of resilience and psychological burden. At the beginning of the BMT and at week 11, 525 male recruits (mean age: 20.3 years) completed a series of questionnaires covering demographic information and assessing resilience, perceived stress and mental distress. In parallel, their superiors rated recruits’ military performance in week 13. Dropout rates were also registered. Cross-sectionally and longitudinally, higher resilience scores predicted lower scores for perceived stress, mental distress, and better military performance. Higher self-rated resilience was moderately associated with military performance, as rated by recruits’ superiors. Resilience scores, perceived stress and mental distress did not differ between those recruits continuing their BMT and dropouts. In support of our assumptions, resilience acted as a protective factor during Swiss Armed Forces BMT.

## 1. Introduction

Stress in the workplace appears to have become one of the most impactful health problems [1]. This holds true for both civilian (private companies; governmental institutions) and military contexts (military personnel in basic military training; military personnel during peace and war missions). Furthermore, while research on stress has focused on the causes and consequences of stress in the workplace [2,3,4,5,6,7,8], research into stress-protective factors is much less extensive [9,10,11]. First, we provide a brief narrative overview of stress and workplace-related stress; we then provide a brief narrative overview of stress among military personnel. Last, we report a study examining whether resilience acts as a protective factor against stress in a sample of Swiss male recruits during their basic military training (BMT).

Stress is understood as “… problems and issues that are either so regular in the enactment of daily roles and activities, or so defined by the nature of daily role enactments or activities, that they behave as if they are continuous for the individual” [12]; p. 82). From a physiological perspective, stress can be defined as a consecutive process consisting of a stimulus (stressor), followed by the activation of a physiological set-up to respond adequately to the stressor (stress response) such as fight, flight or freeze [13]. From the viewpoint of cognitive-emotional information processing, stress is understood as the subjective sense of environmental or internal stimuli as threatening (subjective appraisal of a stimulus as being a stressor). Next, a series of cognitive-emotional strategies (so-called coping strategies) to deal with the stressor have as their objectives decreasing both the stressors and short- and long-term stress reactions [14]. In this respect, while it may be that some stressors, such as the death of a spouse, divorce, being sent to jail, or personal injury are universal [15], retirement, pregnancy, and outstanding personal achievements can also be stressors but to a lesser extent and with substantial differences between people in how they are affected by such events [15].

With regard to workplace-related or occupational stress, this may cause somatic and psychological health problems [2,6,7]; half of all work absences appeared to be associated with stress-related disorders. Absenteeism, employee turnover, and higher rates of accidents have all been causally linked to consequences of work-related stress [3,4,5,16,17].

Given this general background and shifting to the military context, military personnel are exposed to different kinds of stress [18,19]. Almost by definition, people in the military are executing dangerous and stressful work and accordingly, they also have an elevated risk of suffering from stress-related disorders following military deployment [18]. Not surprisingly, and compared to those not in the military, military personnel are at a higher risk of developing other psychological disorders such as depression, anxiety, and alcohol abuse [20]. Approximately 15% of deployed soldiers develop mental disorders, for example, posttraumatic stress disorder [20,21,22]. Therefore, studies of military personnel have focused initially on the development of problems such as psychiatric disorders after combat deployments [23,24,25]. In this respect, and compared to non-military personnel, military personnel have a higher prevalence for one or more of the common psychiatric disorders; such psychiatric disorders have been causally linked to the stressors specific to military service [21,22,26,27]. While not all those in the military are involved in combat settings, as Nakkas et al. [18] observed, military training in peacetime is intended to provide preparation for combat situations. That is to say, the goal of military training is to enable service personnel to perform under adverse conditions characterized by low control, high stress, and high levels of uncertainty.

With regard to the situation in Switzerland, military service is mandatory for all adult males and optional for adult females (https://fedlex.data.admin.ch/filestore/fedlex.data.admin.ch; retrieved 6 April 2021). Therefore, given that military service in Switzerland is not a choice for men, but civic duty, basic military training (BMT) could be perceived as stressful [18,19,28], leading to mental health problems reflected in dropout rates [19]. Minorities have reported higher somatization and less effective coping styles when compared to their non-minority peers [19], while subjectively perceived social support has been associated with more effective coping styles. In contrast, dysfunctional coping styles have been associated with higher scores for depression, somatization, anxiety, interpersonal sensitivity, and hostility. Importantly, those recruits identified as willing and suitable for promotion had more effective coping styles and lower scores for depression, somatization, anxiety, interpersonal sensitivity, and hostility when compared to those unwilling to accept promotion or unsuited to it [18].

Research on stress and coping has shown that not everyone experiencing life stressors develops any form of adverse health. More specifically, research in personality psychology has identified individual differences in adapting to stressful environments [29]. A protective factor against stress appears to be resilience [9,30]. Resilience is understood as a person’s psychological ability to resist and adapt to significant stress, adversity [9,10,11,30,31,32,33,34,35] and trauma [36,37]. Resilience can be further understood as the individual’s ability to achieve, retain, or regain a level of physical or emotional health after illness or loss [38]. Resilience is positively associated with a higher quality of life [39] and negatively associated with vulnerability [40]. Individuals scoring high on resilience are better able to handle major life events and chronic stressors [9,32]. In addition, resilience helps individuals to establish a stable balance in stressful situations [41,42,43], as they recover faster from stress and injury [44,45]. Compared to people scoring low on resilience, individuals scoring high on resilience perform more adequately in the face of safety-threatening events [46,47] and deal successfully with stressful situations [29]. Resilient people have higher scores for self-esteem, self-efficacy, stable interpersonal relationships, and efficient problem-solving [48,49,50].

With regard to resilience in the military context, previous studies have investigated resilience as a protective factor (combat; deployment). Resilience protects military personnel exposed to life-threatening situations in combat and prevents degradation of psychological and physical health [41,42,43]. Longitudinally, dimensions of resilience appear to protect returning military personnel from negative consequences of traumatic exposure [51]. Higher scores for hardiness, a psychological concept substantially overlapping with resilience [52,53,54], predicted better performance among US-Army recruiters [55] and a decreased risk of alcohol abuse [56].

Given this background, improving the resilience of people working in military service may help them avoid mental distress and reduce psychiatric disorders. However, whether resilience has a protective effect on perceived stress, mental distress, performance, or dropout in BMT settings has not been studied so far in the context of the Swiss military. We believe that these questions deserve more attention for the following reasons. First, military service in Switzerland is a civic duty; accordingly, it is unlikely that there is any positive selection bias in BMT towards candidates particularly motivated and willing to undertake military service and thus probably also more resilient. Second, previous studies in the Swiss military context [18,19,28] have shown unfavorable coping styles to be associated with higher psychological distress in the form of depression, anxiety, somatization, and interpersonal sensitivity. Third and relatedly, such associations were observed above all among minorities, dropouts, and recruits not recommended for military promotion. Fourth, it remains unclear thus far whether aspects of resilience could be a protective factor in the long-term by keeping levels of stress low.

Given this background, the aims of the present study were four-fold: First, to investigate the associations between resilience, perceived stress and mental distress at the beginning of the BMT; second, to calculate if and to what extent resilience at baseline could predict perceived stress and mental distress at week 11 of the BMT; third, to investigate if resilience, perceived stress, and mental distress were associated with military performance, as rated by recruits’ superiors; fourth and last, if recruits dropping prematurely from the BMT showed specific psychological profiles, compared to those completing the BMT at least until week 11. Specifically, there is no study that has evaluated the long-term impact of resilience on subjectively perceived stress, mental distress, dropout rate, or performance in BMT of Swiss Army recruits.

The following four hypotheses were formulated. First, based on previous results [3,9,18,19,22,28,30,51], we anticipated that cross-sectionally (at baseline), higher resilience scores would be associated with lower perceived stress and mental distress. Second, we also expected higher resilience scores at baseline to predict lower perceived stress and lower mental distress scores at week 11. Third, following others [55,57,58], we predicted that higher resilience scores at baseline would be associated with the excellence of military performance, as rated by recruits’ superiors. Fourth, following Youssef and Luthans [59], we predicted that recruits completing the first half of their BMT would have higher self-reported resilience, lower perceived stress and lower mental distress scores at baseline compared to those recruits who dropped out of BMT for psychological health or other reasons.

## 2. Methods

### 2.1. Procedure and Study Design

Recruits at the Swiss Armed Forces Infantry School of Aarau (Switzerland) were approached during their BMT to participate in the present online-run study on the longitudinal relation between resilience and psychological functioning. Participants were informed about the aims of the study and the confidential data handling. Specifically, recruits were informed verbally and in writing about the study and that participation and non-participation would have neither a favorable nor unfavorable influence on their BMT. Basic military training lasts 21 weeks. The study design was as follows: The survey was completed at two time points, at the beginning of BMT (baseline) and again at week 11. At baseline, after signing a declaration of consent, participants completed questionnaires covering demographic information, resilience, perceived stress, and mental distress (see details below). Next, at week 13, recruits’ instructing officers (superiors) rated their military performance (see details below). At week 11, some recruits left BMT for physical or psychological health reasons (“dropouts”). The Ethics Committee of the Canton of Aargau (Aarau, Switzerland; AGEK (Arbeitsgemeinschaft der Schweizerischen Forschungs-Ethikkommission für klinische Versuche, Aargau, protocol code 2011/008; date of approval: 8 April 2011)) approved the study, which was performed in accordance with the seventh and current [60] edition of the Declaration of Helsinki.

The study was conducted during autumn 2011 and spring 2012. The present analysis was part of a broader study of biological and psychological characteristics of recruits to the Swiss Armed Forces [28,61,62,63,64,65]. However, the present results have not previously been published and therefore are novel.

### 2.2. Participants

In Switzerland, military service is compulsory for all men. Therefore, all male citizens become liable for conscription at the age of 18. An expert decision as to whether an individual is fit for military service is made in a 2–3-day period of recruitment, and around 65% of those assessed are selected for military service. Specifically, for the following reasons, one is excluded from military service: 1. Chronic medical issues such as severe injuries of the musculoskeletal system, severe diabetes, vision impairment higher than 8 deportees, severe deficiencies of the immune system (e.g., HIV, multiple sclerosis); 2. psychiatric issues such as substance use disorder, severe mood and anxiety disorders, schizophrenia spectrum disorders, antisocial personality disorder; 3. Criminal background, as ascertained from the criminal records. The present study sample is, therefore, representative of physically and mentally healthy young men in Switzerland. The inclusion criteria were: 1. male sex; 2. compliance with the study conditions and specifically understanding German; 3. signed the written informed consent. Exclusion criteria were: clicking through the items either only on the right or left side within a time lapse of some minutes, resulting in a standard deviation of zero on questionnaires with inverse items (called “click-throughs”).

A total of 694 recruits initially agreed to participate. Of these, 43 (6.20%) did not sign the written consent, 81 (11.67%) had French or Italian as their mother tongue, and 45 (6.48%) were identified as “click-throughs”.

The final sample consisted of 525 participants (see Figure 1).

### 2.3. Measures

#### 2.3.1. Demographic Information

Participants reported their age (in years), highest educational level (lower secondary school; upper secondary school; academic high school), and mother tongue (German, French, Italian).

#### 2.3.2. Resilience

Participants completed the German short version [66] of the Resilience Scale (RS-11) (original English version: [48]. The RS-11 scale is a uniform measure of perceived self-confidence/self-efficacy [67]. Typical items are “I usually manage one way or another.”, “I am determined.” and “My life has meaning.” Answers are given on 7-point Likert scales ranging from 1 (=strongly disagree) to 7 (=strongly agree), with higher sum scores reflecting a higher self-rated degree of resilience (Cronbach’s α = 0.92).

#### 2.3.3. Perceived Stress

To assess subjectively perceived stress over the previous four weeks, participants completed the German translation [68] of the perceived stress questionnaire (PSQ) [69]. The dimensions of stress assessed are joy, tension, demands, and worries. Typical items are “You have too many things to do.”, “You feel frustrated.”, and “You are full of energy.” The 20 items are answered on 4-point Likert scales ranging from 1 (=very seldom) to 4 (=almost ever), with some items reverse scored. Higher mean scores reflect greater perceived stress. The values of the subscales and the PSQ Index are mean values. These are derived from the item raw scores and linearly transformed to values between 0 and 1, as recommended in the questionnaire’s manual (Baseline: Cronbach’s α = 0.74; week 11: Cronbach’s α = 0.76).

#### 2.3.4. Mental Distress

To assess subjectively perceived mental distress, participants completed the German version of the Brief Symptom Inventory (BSI) [70], based on the original long version Symptom-Check-List (SCL 90-R) [71]. The following symptoms are assessed over a period covering the past seven days (Somatization (7 items), Obsession-Compulsion (6 items), Interpersonal Sensitivity (4 items), Depression (6 items), Anxiety (6 items), Hostility (5 items), Phobic anxiety (5 items), Paranoid ideation (5 items), and Psychoticism (5 items)). Four additional items have clinical importance but do not form part of any of the subscales. Typical items are “unpleasant thoughts,” “nervousness,” and “restless sleep.” Answers are given on 5-point Likert scales ranging from 0 (=not at all) to 4 (=extremely), with higher sum scores reflecting a more pronounced degree of mental distress. In addition, a global index of distress is calculated, the global severity index (GSI). (Baseline: Cronbach’s α = 0.95; week 11: Cronbach’s α = 0.98).

#### 2.3.5. Military Performance

To assess recruits’ military performance, instructors completed the military performance form [72]. This form covers the evaluation of the recruit’s behavior and performance in 2 categories each with 2 subscales. The first category is self and social capacity, with the subscales personal attitude and social behavior. The second category, labelled military expertise, is composed of the subcategories military performance and outcomes of inspections, exams, and physical performance tests. All scales are based on a 5-point Likert scale ranging from 1 (=insufficient) to 5 (=excellent), with a higher overall mean score reflecting a better military performance (Cronbach’s α = 0.81).

#### 2.3.6. Dropouts

Recruits who resigned from BMT were classified as dropouts. Dropouts are grouped into two different categories: dropouts due to mental health or dropouts due to other (physical health, accidents, other reasons).

### 2.4. Statistical Analysis

Preliminary information: Given the naturalistic and rather explorative character of the present study, which has been conducted for the first time in a Swiss Army setting, no sample size calculation was performed.

Next, we performed preliminary calculations with age, educational level and BMI as possible confounders; none of these variables did systematically change or biased scores of resilience, perceived stress, mental distress and military performance. Given this, the decision was to keep the current pattern of statistical procedure unaltered.

Pearson’s correlations were computed between age, resilience at baseline, perceived stress and mental distress both at baseline and at week 11, and military performance at week 13.

The residuals of the regression models were tested for normality. To predict perceived stress at week 11 of BMT, hierarchical linear regression was performed. Resilience, perceived stress, and mental distress at baseline were predictors with perceived stress during week 11 as the dependent variable. The same procedure was adopted to predict mental distress at week 11 and military performance at week 13 of BMT.

For group comparisons, a multivariate ANOVA was performed. The mental health of the dropout group, the group of dropouts for other reasons, and the non-dropout group were compared with respect to resilience, perceived stress, and mental distress at baseline.

The level of significance was set at *p* < 0.05 for all calculations. Note that during the BMT, some participants dropped out from military service, and due to duty-related constraints, not all participants were able to complete all questionnaires. Given this, there were variations in sample size across analyses.

Statistical analyses were conducted using IBM SPSS Statistics 21 (IBM Corporation, Armonk, NY, USA).

## 3. Results

### 3.1. Sample Characteristics

Participants were all males (*N* = 525; mean age: 20.3 years (SD = 1.16); BMI (mean BMI: 23.51 (SD = 3.06)). Of these, 163 (31%) had completed lower secondary school, 203 (38.7%) had completed upper secondary school, and 159 (30.2%) had completed academic high school.

### 3.2. Resilience, Perceived Stress and Mental Distress at Baseline and at Week 11, and Military Performance at Week 13

Table 1 provides the descriptive statistics and the correlations between resilience, perceived stress, and mental distress at baseline and at week 11 and military performance at week 13, as rated by recruits’ superiors.

Higher scores for resilience at baseline were associated with lower perceived stress and mental distress both at baseline and at week 11. Additionally, higher scores for resilience at baseline were associated with the excellence of military performance at week 13, as rated by recruits’ superiors.

Higher perceived stress at baseline was associated with higher perceived stress at week 11, with higher mental distress at baseline and at week 11, and with a poorer military performance at week 13.

Higher mental distress at baseline was associated with higher perceived stress and higher mental distress at week 11, while mental distress was not associated with military performance at week 13.

Higher perceived stress at week 11 was associated with higher mental distress at week 11, while perceived stress at week 11 was not associated with military performance at week 13.

Similarly, higher mental distress at week 11 was not associated with military performance at week 13.

Overall, higher resilience at baseline was associated with lower perceived stress, lower mental health both at baseline and at week 11, and with the quality of military performance at week 13 as rated by recruits’ superiors.

### 3.3. Resilience at Baseline as Predictor of Perceived Stress and Mental Distress at Week 11 and Military Performance at Week 13

To calculate the influence of resilience at baseline on perceived stress and mental distress at week 11, a series of hierarchical regression analyses were performed.

### 3.4. Perceived Stress at Week 11

Table 2 provides the statistical overview of the long-term influence of resilience (baseline) on subjectively perceived stress at week 11, controlling for subjectively perceived stress and mental distress at baseline.

Both higher perceived stress and mental distress at baseline and lower resilience predicted higher perceived stress at week 11. Adding resilience to the model (step 2) increased the strength of the model to a modest but significant degree.

### 3.5. Mental Distress at Week 11

A hierarchical regression analysis was conducted to test the long-term influence of resilience (baseline) on mental distress at week 11, controlling for baseline values of mental distress and subjectively perceived stress in a first step. Table 3 provides the statistical overview. Both higher mental distress and lower resilience at baseline predicted higher mental distress at week 11. Perceived stress at baseline was unrelated to mental distress at week 11. Adding resilience to the model (step 2) increased the strength of the model to a modest but significant extent.

### 3.6. Military Performance at Week 13

A hierarchical regression analysis was conducted to test the long-term influence of resilience (baseline) on military performance at week 13, controlling for baseline values of perceived stress and mental distress in a first step. Table 4 provides the statistical overview. Higher resilience scores at baseline predicted military performance scores at week 13, while perceived stress and mental distress at baseline were unrelated to military performance. Adding resilience to the model (step 2) increased the strength of the model to a modest but significant degree.

### 3.7. Resilience, Perceived stress and Mental Distress at Baseline among Recruits Continuing Their Military Duty and Dropouts

Table 5 provides the descriptive and inferential statistical overview of resilience, perceived stress and mental stress between recruits continuing their military duty and recruits who left the BMT early due to psychological or other reasons (dropouts).

All effect sizes were small. Descriptively, dropouts for psychological reasons had medium scores for resilience and the highest perceived stress and mental distress scores (always at baseline).

## 4. Discussion

The key findings of the present study of young adult Swiss males performing their civic duty as recruits during basic military training (BMT) were as follows. Higher self-rated resilience at baseline predicted lower self-perceived stress and lower mental distress, both cross-sectionally (baseline) and longitudinally at week 11, and quality of military performance at week 13, as rated by recruits’ superiors. In addition, recruits leaving BMT before completion (dropouts) had more negative scores for resilience, perceived stress and mental distress, though the effect sizes were small. The novelty of the present study is that findings add to the current literature in three ways. First, it appeared that resilience was a protective factor for perceived stress and mental distress longitudinally. Second, this pattern of results was observed during BMT among Swiss recruits, who by law are required to discharge this military duty. Third, superiors responsible for recruits’ BMT perceived those recruits with higher self-rated resilience scores to be better able and willing to comply with their military duty.

Four hypotheses were formulated, and each of these is considered now in turn.

Our first hypothesis was that at baseline, recruits with higher self-rated resilience scores would also report lower perceived stress and lower mental distress, and this was confirmed. In this respect, the present findings are in accord with previously reported results [3,9,18,19,22,28,30,39,45,51,73], and our results can be taken as further confirmation and replication of what appears to be plausible. However, the present findings expand upon previous work in that these patterns were observed among Swiss male recruits during their BMT.

Our second hypothesis was that higher resilience scores would predict lower perceived stress and lower mental distress at week 11, and again this was supported. Accordingly, we were again able to replicate what has been observed outside of a military context [9,10,11,31,32,33,34,35]. Again, however, the present findings expand upon previous studies in two ways. First, these links have not previously been reported for Swiss recruits undertaking their BMT, who by law are obliged to discharge this civic duty. Therefore, we assume that motivation was not so high as to have biased the pattern of results. Second, we note that most of the earlier studies in a military context have employed purely cross-sectional designs [56,74,75,76]. In contrast, the present design was longitudinal. As an additional point, it would be interesting to know if these results could be replicated in a civilian environment to indicate that higher resilience can predict better coping with stress in the long-term.

Our third hypothesis was that higher resilience scores at baseline would be associated with superiors’ ratings of excellence of military performance at week 13, and this was supported. We believe that in this respect, our results mirror previous findings [55,56,57,58]. We note that the correlation coefficient was modest (r = 0.15); to put it another way, variance in resilience predicted 2.25% of the variance in rated military performance. Nevertheless, for the following two reasons, we see this finding to be of importance. First, the correlation was in the predicted direction. Second, this was a comparison of self-ratings and superiors’ ratings, and by nature, a high degree of overlap could not be assumed. Indeed, even the overlap in symptoms of depression rated, respectively, by patients and experts is modest. Given this, this observation holds even more true given that a self-rated attitude of self-confidence and self-efficacy is compared with dimensions of military performance such as self and social capacity, along with performance and outcomes of inspections, exams and physical performance tests [72].

The data available from this study is unable directly to clarify the underlying psychological mechanisms and particularly the “transmission belts” turning a psychological construct such as resilience into behavior perceived by the recruit’s responsible officer as excellence in military performance. Given these issues, we offer the following admittedly highly speculative possibilities.

First, resilience is understood as the psychological ability to resist and adapt to significant stress and adversity [3,9,18,19,22,28,30,39,44,51,73]. Accordingly, one might expect that a recruit with high resilience is able and willing to cope with stress and the physical and psychological challenges of BMT (e.g., long working hours, sleep deprivation, limited time for personal use, etc.). Second, at a behavioral level, this kind of ability and willingness should be observable in the form of adaptation to the constraints and requirements of military life. Third, one can assume that resilient recruits will be motivated to learn, adapt and perform in a military environment while also retaining a balance between personal aims and social behavior towards the other recruits with whom they must share their daily lives. Fourth, the time gap of 12 weeks between the self-assessment of resilience and the assessment of military performance as rated by the recruits’ superiors is worthy of note. Though again highly speculative, it is possible that the modest correlation reflected changes in unassessed factors such as injuries, disappointments, alterations of attitudes towards military duty and military life. Overall, though the association between resilience and military performance was modest, it was in the expected direction.

Our fourth hypothesis was that recruits unable to continue their BMT (dropouts) would have lower scores for resilience and higher scores for perceived stress and mental distress, though this was not supported (see Table 5). The present results are, therefore, in contrast to those reported by Youssef and Luthans [59]. Importantly, recruits unable to complete their military duty for mental health reasons had descriptively higher scores on perceived stress and mental distress at baseline, but descriptively higher resilience scores, than dropouts for physical reasons though these differences were not statistically significant. Given this, resilience scores at baseline should not be used as predictors of future dropouts.

Despite the replication and novelty of the present results, the following limitations warrant against overgeneralization. First, we assessed only a small number of male recruits carrying out their military duty; accordingly, sample biases cannot be ruled out. This further means that the generalizability of the present results might be modest. Second, we assessed exclusively male recruits, though, unlike in other armies, such as the US army, the percentage of female soldiers in the Swiss Army is very low. Third, the modest to medium correlation coefficients imply that larger portions of the variance in perceived stress, mental distress and military performance remain unexplained. Therefore, it is entirely possible that latent and unassessed psychological dimensions might have biased two or more variables in the same or opposite directions. Specifically, given that military service is compulsory for all male Swiss adults, it is also highly conceivable that motivational issues (willingness to serve) might have blurred the current pattern of results. Likewise, it is conceivable that further unassessed variables, such as bullying [77], belonging to an ethnic minority [19] or the risk of unemployment after the military service might have unfavorably impacted the present pattern of results. Fourth, we basically relied on self-reports; this might be the reason for the modest overlap between the self-reported dimensions and military performance, as ratted by their superiors. Fifth, with regard to the prediction of military performance, the results of the regression models yielded modest associations (R^2^); these results should not, therefore, be overstated. Sixth, the present data cover the first 13 weeks of BMT; it would be informative to know if the variables we assessed have any predictive value by completion of BMT. Seventh, while the present pattern of results offers a nice model for explaining the impact of resilience on stress coping longitudinally among Swiss recruits, it would be interesting to explore the applicability of this model to a civilian working environment. Eighth, while, for instance, the US Army offers specific resilience training programs [73,78] to their service men with encouraging results, such specific interventions are still lacking in the Swiss Army.

Future studies among BMT of Swiss Armed Forces might address resilience training programs [73,78] and other important issues for the Army as bullying [77] or willingness to serve in military service. Other topics of interest could be the role of minorities, somatization levels and coping styles [18,19] as well as predictors and personal improvements.

As it was shown before, that online CBT interventions have a positive impact on depression, anxiety [79,80,81], and insomnia [82], further studies might include the survey of online-CBT interventions and their impact on diverse psychological variables in a military context, for example during BMT.

## 5. Conclusions

Among a small sample of Swiss male recruits undertaking their compulsory BMT, resilience appeared to be a valuable psychological construct associated with lower perceived stress and lower mental distress both cross-sectionally and longitudinally at week 11 of the training, which is the mid-point of Swiss BMT. Furthermore, self-rated resilience was associated with superior-rated excellence of military performance at week 13.

## Figures and Tables

**Figure 1 ijerph-18-06077-f001:**
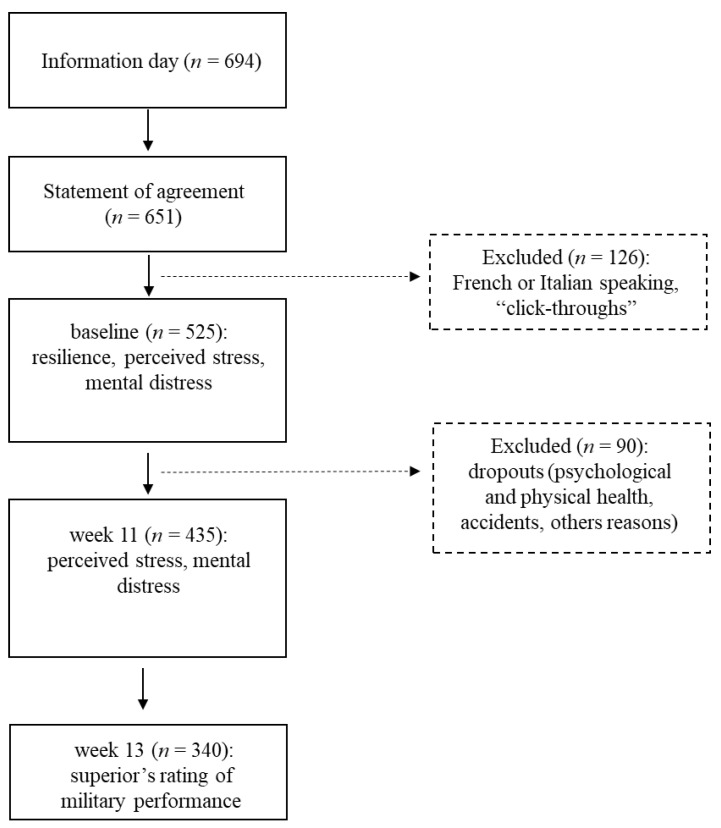
The flow chart shows the sample size of participants. In two examinations (baseline and at week 11), self-rating data covering demographic information and psychological dimensions of resilience, perceived stress, and mental distress were collected. In week 13, an expert rating was made of the military performance of the recruits.

**Table 1 ijerph-18-06077-t001:** A descriptive and correlational overview of resilience at baseline, perceived stress and mental distress at baseline and week 11, and military performance at week 13.

Variables			Timepoints							
	Baseline				Week 11			Week 13	M	SD
	Resilience	Perceived Stress		Mental Distress	Perceived Stress	Mental Distress	Military Performance			
Sample size (*n*)	459	520		517	361	352	340			
Baseline										
Resilience	-	−0.38 **		−0.27 **	−0.28 **	−0.24 **	0.15 *		58.9	11.1
Perceived stress		-		0.55 **	0.41 **	0.12 *	−0.11		32.2	16.5
Mental distress baseline				-	0.33 **	0.24 **	−0.05		0.47	0.45
Week 11										
Perceived stress					-	0.30 **	−0.11		41.5	14.4
Mental distress						-	0.00		0.58	0.66
Week 13										
Military performance							-		2.82	0.46

Notes. * *p* < 0.05, ** *p* < 0.01; M: mean; SD: standard deviation.

**Table 2 ijerph-18-06077-t002:** The influence of resilience (baseline) on perceived stress (week 11) in a hierarchical regression analysis, controlling for baseline values of perceived stress (baseline) and mental distress (baseline).

	Perceived Stress (Week 11)				
Variables	*B*	*SE B*	*β*	R^2^	ΔR^2^
Step 1					
Perceived stress (baseline)	0.28	0.05	0.32 ***	0.20	0.19
Mental distress (baseline)	6.70	2.04	0.20 **		
Step 2					
Perceived stress (baseline)	0.24	0.06	0.27 ***	0.21	0.20
Mental distress (baseline)	6.26	2.03	0.18 **		
Resilience (baseline)	−0.16	0.08	−0.12 *		

Notes. N = 300. * *p* < 0.05, ** *p* < 0.01, *** *p* < 0.001; B: unstandardized beta; SE B: standard error for the unstandardized beta; β: standardized beta; R^2^: R-squared; ΔR^2^: Delta R-squared.

**Table 3 ijerph-18-06077-t003:** The influence of resilience (baseline) on mental distress at week 11 in a hierarchical regression analysis, controlling for baseline values of mental distress GSI and perceived stress.

	Mental Distress (Week 11)				
Variables	*B*	*SE B*	*β*	R^2^	ΔR^2^
Step 1					
Mental distress (baseline)	0.41	0.10	0.26 ***	0.07	0.07
Perceived stress (baseline)	0.00	0.00	0.01		
Step 2					
Mental distress (baseline)	0.38	0.10	0.24 ***	0.10	0.09
Perceived stress (baseline)	0.00	0.00	−0.06		
Resilience (baseline)	−0.01	0.00	−0.19 **		

Notes. N = 294. ** *p* < 0.01, *** *p* < 0.001; B: unstandardized beta; SE B: standard error for the unstandardized beta; β: standardized beta; R^2^: R-squared; ΔR^2^: Delta R-squared.

**Table 4 ijerph-18-06077-t004:** The influence of resilience (baseline) on military performance at week 13 in a hierarchical regression analysis, controlling for baseline values of perceived stress and mental distress.

	Military Performance (Week 13)				
Variables	*B*	*SE B*	*β*	R^2^	ΔR^2^
Step 1					
Perceived stress (baseline)	0.00	0.00	−0.10	0.01	0.00
Mental distress (baseline)	0.02	0.08	0.02		
Step 2					
Perceived stress (baseline)	0.00	0.00	−0.06	0.03	0.01
Mental distress (baseline)	0.02	0.08	0.02		
Resilience (baseline)	0.01	0.00	0.13 *		

Notes. N = 275. * *p* < 0.05; B: unstandardized beta; SE B: standard error for the unstandardized beta; β: standardized beta; R^2^: R-squared; ΔR^2^: Delta R-squared.

**Table 5 ijerph-18-06077-t005:** Multivariate ANOVA for group comparisons of the mental health dropout group, the group of dropouts for other reasons, and the non-dropout group of variables resilience, perceived stress and mental distress at baseline.

		Group		
	Dropout:	Dropout Other Reasons:	Non-Dropouts	Group
	Mental Health	Physical Health, Accidents, Others		
Degrees of freedom				2
N	20	50	384	
	M (SD)	M (SD)	M (SD)	*F* partial eta^2^
Resilience (baseline)	57.00 (11.24)	55.54 (12.48)	59.48 (10.87)	3.13 * 0.01 [S]
Perceived stress (baseline)	46.75 (22.74)	35.97 (17.10)	31.40 (15.80)	9.64 *** 0.04 [S]
Mental distress (baseline)	0.85 (0.71)	0.58 (0.52)	0.44 (0.41)	9.48 *** 0.04 [S]

Notes. [S] = small effect size; * *p* < 0.05, *** *p* < 0.001; M: mean; SD: standard deviation; F: F-ratio.

## Data Availability

Data belong to the Swiss Armed Forces and to the Swiss Federal Office of Sport; data are made available to experts in the field upon request and upon the detailed description of the reason of request.

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
