# Peer review of "Resilience as a Protective Factor in Basic Military Training, a Longitudinal Study of the Swiss Armed Forces"

_ijerph, 2021, doi:10.3390/ijerph18116077_

Round 1
Reviewer 1 Report
Thank you for allowing me to review this manuscript. This manuscript entitled "Resilience as a protective factor in basic military training, a longitudinal study 2 of the Swiss Armed Forces". The present study examined whether resilience would act as a protective factor 24 during BMT in the Swiss Armed Forces.
It is an interesting project, with a very current theme, although it has several limitations that make it susceptible to publication in this magazine. These limitations are detailed below:
- In relation to the citation regulations, there are errors in the order of the citations. It is necessary to review the citations so that they comply with the regulations indicated by the journal.
- At the end of the introduction, the objective of the article is not detailed. It would be important for them to be reflected in the last paragraph of the section.
- The material and methods section is very complete, and the main data that should be included in this section are identified. However, the study design is not clearly specified.
- In the results section, the tables do not specify a footer for those used in them.
- The conclusions are clear and precise, but it would be interesting to include some future lines due to the importance and timeliness of the subject.
- As noted, the central theme of the manuscript is highly topical, which requires that the bibliography be from recent years. However, in the bibliographic references section, there are articles with more than 10 years. It would be necessary to review this aspect and eliminate those citations that are not up to date.
Author Response
We thank Reviewer #1 for the care devoted to improve the quality of the manuscript. Please find the detailed point-by-point-response attached as a separate file. Thank you again for all your kind efforts.

Reviewer 2 Report
The overall small effect sizes are concerning, especially given the lack of adjusting for subject characteristics. In general, many prior studies have found that higher resilience is associated with better mental health, reduced stress, and greater well-being, hence the results are somewhat expected and lack novelty.
Specific comments:
- Please change "As regards" to "With regard to".
- There needs to be a better and more varied writing style. The authors keep using the same transition between paragraphs, "As regards..."
- Based on my understanding, the Swiss government imposes the duty to serve in the military on every Swiss male, but allows conscientious objectors to enlist in a compulsory civil service?
- How was the sample size determined? There is currently no evidence of power calculation.
- Please provide the actual IRB study/approval number.
- Was the study protocol prospectively registered?
- The effect sizes were all small. They were also unadjusted for potential covariates. More elaboration and exploration are necessary.
- There were other study limitations, e.g. use of self-reported outcomes, and the lack of demographic diversity limits the generalizability of the study findings.
- How can we encourage and support psychological resilience to stress for these servicemen? At least some comments are necessary.
- What were some of the reasons for dropout from the study?
- The underlying data should be made publicly available. If this was not possible, please provide a reason why.
Author Response
We thank Reviewer #2 for the care devoted to improve the quality of the manuscript. Please find the detailed point-by-point-response attached as a separate file. Thank you again for all your kind efforts.

Reviewer 3 Report
I have the following comments for the authors to address. I am happy to review this paper again.
1) The authors stated " Swiss male recruits during their basic military
training (BMT)" Can the authors provide more information. Is BMT compulsory in Switzerland? Are these men voluntary or involuntary?
2) The authors should state the duration of BMT.
3) Under the method, the authors should state the recruitment period. Was it during the COVID-19 pandemic?
4) The authors should state the following limitations. There are other confounding factors that this study did not address. First, this study did not assess bullying that is very common in the army. Second, this study did not assess for physical fitness e.g BMI or obesity, the higher BMI, the higher stress level. Third, the authors should assess willingness to serve BMT. Lastly, the authors did not provide any information of chronic medical illness.
5) The authors should discuss future study such as resilience and burnout of military personnel.
6) The authors should propose psychological intervention. The authors should mention digital cognitive behavior therapy that can treat psychiatric symptoms such as insomnia that is common in military personnel.. Please search Pubmed for the studies that state "Although cognitive behavioural therapy for insomnia (CBT-I) has been recommended the initial therapy for insomnia, its clinical usage remains limited due to the lack of therapists. Digital CBT-I (dCBT-I) can potentially circumvent this problem"
Author Response
We thank Reviewer #3 for the care devoted to improve the quality of the manuscript. Please find the detailed point-by-point-response attached as a separate file. Thank you again for all your kind efforts.

Round 2
Reviewer 2 Report
Thank you for the revisions.
Specific comments:
- As per the journal's guidelines, the abstract should be a total of about 200 words maximum and without headings.
- The citations were not numbered in running order. Please check.
- "This further means that the results not generalizability is modest" - what does this mean? The manuscript is in general need of wordsmithing.
- "Fifth, relied on effect sizes; mean differences in resilience, perceived stress and mental health among dropouts and non-dropouts were modest" - do you mean the effect sizes were overall small?
- "... military service is compulsory for all male Swiss adults" - are medical exemptions granted on grounds of serious psychiatric illness?
Author Response
Again, we thank Reviewer #2 for drawing our attention to further issues of the manuscript. We have addressed these issues; please find the detailed point-by-point-response attached as a separate file. Again, thank you very much for the care devoted to thoroughly review the manuscript.
